# Induction of pro-inflammatory genes by fibronectin DAMPs in three fibroblast cell lines: Role of TAK1 and MAP kinases

**Pranav Maddali, Anthony Ambesi, Paula J. McKeown-Longo** *

Department of Regenerative & Cancer Cell Biology, Albany Medical College, Albany, New York, United States of America

* mckeowp@amc.edu

**Data Availability Statement:** All relevant data are within the paper and its supporting information files

## Abstract

Changes in the organization and structure of the fibronectin matrix are believed to contribute to dysregulated wound healing and subsequent tissue inflammation and tissue fibrosis. These changes include an increase in the EDA isoform of fibronectin as well as the mechanical unfolding of fibronectin type III domains. In previous studies using embryonic foreskin fibroblasts, we have shown that fibronectin's EDA domain (FnEDA) and the partially unfolded first Type III domain (FnIII-1c) function as Damage Associated Molecular Pattern (DAMP) molecules to stimulate the induction of inflammatory cytokines by serving as agonists for Toll-Like Receptor-4 (TLR4). However, the role of signaling molecules downstream of TLR-4 such as TGF-β Activated Kinase 1 (TAK1) and Mitogen activated protein kinases (MAPK) in regulating the expression of fibronectin DAMP induced inflammatory genes in specific cell types is not known. In the current study, we evaluate the molecular steps regulating the fibronectin driven induction of inflammatory genes in three human fibroblast cell lines: embryonic foreskin, adult dermal, and adult kidney. The fibronectin derived DAMPs each induce the phosphorylation and activation of TAK1 which results in the activation of two downstream signaling arms, IKK/NF-κB and MAPK. Using the specific inhibitor 5Z-(7)-Oxozeanol as well as siRNA, we show TAK1 to be a crucial signaling mediator in the release of cytokines in response to fibronectin DAMPs in all three cell types. Finally, we show that FnEDA and FnIII-1c induce several pro-inflammatory cytokines whose expression is dependent on both TAK1 and JNK MAPK and highlight cell-type specific differences in the gene-expression profiles of the fibroblast cell-lines.

## Introduction

The extracellular matrix (ECM) consists of a complex assembly of proteins which provide structural and mechanical support to tissues. Fibroblasts are the primary cell type which assembles, maintains, and remodels the ECM to meet the changing needs of the tissue [1]. Dysfunction of this process leads to the development of chronic inflammation and fibrosis [2, 3]. Fibronectin is one of the most abundant components of the ECM and plays a critical role

**Funding:** NIH 1R1AR067956-01A1 awarded to Dr. McKeown-Longo. The Muntz Fund and the Burke Foundations are Institutional awards to Dr. McKeown-Longo. The funders had no role in study design, data collection and analysis.

**Competing interests:** The authors have declared that no competing interests exist.

in regulating cellular behavior. Fibronectin is a 450 kD dimeric, multi-modular glycoprotein, which is polymerized by fibroblast cells into a network of fibers to support cell adhesion, migration, and survival. The fibronectin matrix also serves as a scaffold to provide binding sites for additional ECM molecules, cytokines and growth factors [4–6]. The secondary structure of the fibronectin molecule is organized into three repeating, individually folded domains, Type I, II, and III [7]. The Type I and II domains are stabilized by disulfide bonds, while the Type III domains unfold in response to mechanical force [8]. This conformational lability of the fibronectin molecule allows fibronectin to function as a mechano-sensor regulating exposure of bioactive sites within the matrix [9]. These mechanically-controlled sites have been shown to regulate several cellular processes including cell adhesion [10, 11], integrin signaling [12], fibronectin fiber polymerization [13–15], and the binding of growth factors and cytokines [16–18].

Recent studies have indicated that ECM molecules are an integral part of the innate immune system. Components of the ECM are now recognized as DAMPs [19–22]. Most ECM derived DAMPs work through Toll-like receptors, typically TLR2 or TLR4, to initiate the expression of profibrotic and proinflammatory genes [23, 24]. The alternatively spliced isoform of fibronectin contains an extra Type III domain, extra domain A (EDA), and functions as a DAMP. The synthesis of the EDA isoform is upregulated in fibrotic, inflamed, and diseased tissue, where it serves as an agonist for TLR4 and drives the expression of chemokines, cytokines and ECM remodeling genes [25–28].We have also identified a second Type III domain in fibronectin which serves as an agonist for TLR4 and elicits an innate immune response in human skin fibroblasts cells [29, 30]. This DAMP activity is cryptic within the carboxyl two thirds of the III-1 domain (termed FnIII-1c) and is predicted to be unmasked when the Type III-1 domain, is either unfolded by increased mechanical force or released from the fiber by metalloproteases [31–33]. Although many ECM derived DAMPs work through TLR4 or TLR2, their downstream signaling pathways and specific pattern of gene induction can differ [23]. The molecular basis for these differences is not well understood as most of what is known about the signaling pathway downstream of TLR4 is based on studies using the prototypic TLR4 ligand, Lipopolysaccharide, (LPS). LPS, a component of the cell wall of gram-negative bacteria, is a Pathogen Associated Molecular Pattern Molecule (PAMP) which activates TLR4 signaling to elicit an innate immune response to invading pathogens [34].

Chronic fibro-inflammatory responses can occur in most tissues. Recent studies have shown that the EDA isoform of fibronectin is upregulated in several chronic skin conditions including hypertrophic and keloid scars [35, 36], psoriasis [25, 37–39], and scleroderma [26] where it is thought to promote cutaneous fibrosis by serving as an agonist for TLR4. Biopsies from patients with renal disease exhibit increased amounts of the EDA isoform [40]. Increased EDA fibronectin is also seen in animal models of renal interstitial fibrosis [41]. Preclinical models of renal ischemia/reperfusion injury have shown that the subsequent induction of inflammatory genes is controlled by TLR4 [42]. While it is well established that the TLR4 regulated innate immune response is mediated by NF-κB activation [43, 44], the role of TAK1 in the activation of both NF-κB and the MAPKs is less well understood, as TAK1 has been reported to regulate both pro- and anti-fibro-inflammatory phenotypes [45–51]. Transcription of fibro-inflammatory genes is known to be regulated by both activation of NF-κB and MAPK. However, there are few studies which address the question of cell type specificity in the response of resident stromal fibroblasts to ECM derived DAMPs.

In the current study, we evaluated the role of downstream modulators of the TLR4 signaling pathway in the fibronectin DAMP mediated induction of inflammatory genes in three human fibroblast cell lines: embryonic foreskin, adult dermal, and adult kidney. We found that both FnEDA and FnIII-1c induced the release of the pro-inflammatory cytokine IL-8 in all three

cell types, while simultaneous addition of the fibronectin DAMPs resulted in a synergistic release of IL-8. Earlier studies have reported increased levels of IL-8 in association with chronic inflammatory diseases of skin and kidney [52, 53]. Both FnEDA and FnIII-1c individually activated TAK1, IKK/NF-κB, and the ERK, JNK and p38 MAPKs in the three cell types. Additionally, the fibronectin DAMP induced IL-8 release was dependent on TAK1 activation. Interestingly, while the activation profile of the signaling mediators was similar in the three cell lines, we found differences in fibronectin DAMP induced pro-inflammatory gene expression among the three cell lines, highlighting cell type specific responses to Fn-DAMPs. The fibronectin DAMP initiated induction of cell type specific pro-inflammatory genes may provide avenues for targeted therapies for the treatment of organ specific fibro-inflammatory conditions.

## Materials and methods

### Reagents and antibodies

All antibodies were purchased from Cell Signaling (Danvers, MA, USA). Phospho(T184/187)-TAK1 (#4508), TAK1 (#4505), phospho(S536)-NF-κB(p65subunit) (#3033), phospho(S176/180)-IKKα/β (#2697), phospho(T202/Y204)-ERK (#9106), phospho(T180/Y182)-p38 (#9211), phospho(T183/Y185)-JNK (#9255), GAPDH (#2118), FAK (#13009). The following small molecule inhibitors were used in this study: (5Z)-7-Oxozeanol (Tocris, Bristol, U.K.; 3604), (5Z)-Zeanol (EMDMillipore, Burlington, MA, USA; 499609), Temuterkib (LY3214996) (Selleckchem, Houston, TX, USA; S8534), JNK IX (Santa Cruz Biotech, Dallas, TX, USA; sc-202670), SB202190 (Biomol, Germany). Specific concentrations are mentioned in Figure legends. Recombinant FnEDA and FnIII-1c peptides were prepared and purified as previously described [54, 55].

### Cell culture and treatments

Human foreskin fibroblasts (A1F) [29] and adult human dermal fibroblasts (HDF-ATCC, Manassas, VA-#PCS201-012) were grown and maintained in complete medium (Dulbecco's Modified Eagle's Medium (DMEM; Invitrogen/Life Technologies, Corp., Grand Island, NY, USA) containing 10% fetal bovine serum (FBS; HyClone Laboratories, Logan, UT, USA) supplemented with Pen-Strep (Gibco, Waltham, MA, USA) and GlutaMAX (Gibco) in a 8% $CO_2$ humidified atmosphere at 37˚C. Human Kidney fibroblasts (HKF, gift from Dr. Paul Higgins, Albany Medical College, Albany, NY, USA) were maintained in DMEM GlutaMAX containing 10% FBS supplemented with Pen-Strep in 5% $CO_2$ humidified atmosphere at 37˚C. Prior to all treatments, confluent cell monolayers were rinsed with serum-free medium (DMEM supplemented with 0.1% Bovine Serum Albumin (BSA, Roche Applied Science, Indianapolis, IN, USA), Pen-Strep, GlutaMAX, 1x Non-Essential Amino Acids (NEAA; Gibco) and 10 mM HEPES (Gibco). To obtain whole cell lysates, cells were placed on ice immediately post-treatment, rinsed once with ice cold phosphate-buffered saline (PBS) containing 1 mM sodium ortho-vanadate and were directly lysed in SDS-containing Sample Buffer (62.5 mM Tris, pH 6.8, 2% SDS, 10% glycerol, 50 mM dithiothreitol, and 0.01% bromophenol blue) supplemented with Complete Protease Inhibitor (Roche, Basel, Switzerland) and 10 nM Calyculin A (Sigma, St. Louis, MO, USA; C5552). Samples were denatured immediately by boiling at 100˚C and stored at -20˚C until prior to use.

## Western blot analysis for phosphorylated TAK1

To evaluate TAK1 activation (Thr184/187), cells were lysed in Sample Buffer and boiled for 5 min before separation by electrophoresis on 10% SDS-polyacrylamide gels. Proteins were transferred onto nitrocellulose membranes (GE Healthcare, Uppsala, Sweden) followed by blocking in TBS-T (Tris-HCl, pH 7.4, 150 mM NaCl, and 0.1% Tween 20) supplemented with 5% w/v BSA for 1 h. The membranes were then incubated overnight with a primary antibody to phosphoTAK1 at 4°C on a rocker. The membranes were then rinsed with TBS-T and secondary antibody conjugated with horseradish peroxidase (HRP) was added to detect protein with an enhanced chemiluminescence reagent (Bio-Rad Laboratories, Hercules, CA, USA) in an imager using Image Lab software (Bio-Rad). Densitometric analysis was performed using ImageJ software (National Institutes of Health, Bethesda, MD) by normalizing the phospho-TAK1 signal intensity to GAPDH (loading control).

## Protein analysis using Protein Simple WES system

Cell lysates were analyzed using the automated capillary-based western system (WES) (ProteinSimple, San Jose, CA, USA) as per manufacturer's recommended procedure. For analysis, the signal intensity (area under peak) of the protein of interest was normalized to intensity of the loading control (GAPDH or FAK) using the Compass software (Protein Simple). Additional normalization to experimental controls were performed as stated in Figure legends.

## siRNA-mediated TAK1 knock-down

Small interfering RNAs (siRNAs) (Dharmacon SMARTpool L-003790-00-0010 (TAK1), D-001810-02-20 (Non-Targeting Control #2) were purchased from Horizon Discovery Ltd. (Lafayette, CO, USA). Cells were seeded at a density of 20,000/well in 24-well tissue-culture plates in triplicate in complete medium without antibiotics overnight. The cells were then rinsed with Opti-MEM (ThermoFisher Scientific, Waltham, MA, USA) and transfected with siRNAs to a final concentration of 40 nM using DharmaFECT 2 transfection reagent (Horizon Discovery, T-2005-01) in 400 μl Opti-MEM for 4 hours, followed by addition of 400 μl of DMEM supplemented with 20% FBS-DMEM without antibiotics and left for 5 days. Cells were rinsed with serum-free medium (w/o Pen-Strep) and treated with Fn-DAMPs (as described in Figure legends) for 4 h. Conditioned medium was collected and IL-8 concentrations measured by ELISA. TAK1 knockdown was confirmed using the automated WES system. Cell viability was assessed using a toluidine blue assay as previously described [56].

## ELISA

Cells were seeded at a density of 40,000/well in 48-well tissue-culture plates overnight in complete medium. Cells were treated with fibronectin DAMPs in serum free medium. After 4 h, the conditioned medium was collected and analyzed for IL-8 protein secretion using a human Enzyme-Linked Immunosorbent Assay (ELISA) kit (BD Biosciences, San Diego, CA, USA), as per manufacturer's recommended procedure.

## RNA isolation for real-time RT-PCR

To obtain RNA, $5x10^4$ cells were seeded in 12-well tissue-culture plates overnight in complete medium. Cells were then rinsed with serum-free medium and treated with DMSO or inhibitors for 1 h prior to addition of PBS or fibronectin DAMPs for 3 h (specific concentrations provided in figure legends). Post-treatment, cells were rinsed once with PBS and total RNA was isolated using RNeasy RNA extraction kit (Qiagen, Valencia, CA, USA). An RT² First

Strand kit (Qiagen) was used to convert 0.5 ng of RNA into cDNA as per manufacturer's recommended procedure. The cDNA was added to a RT$^2$ Profiler Human Inflammatory Response and Autoimmunity PCR Array (Qiagen, PAHS-077Z) and real-time PCR analysis was performed in a thermocycler (Bio-Rad Laboratories, Hercules, CA, USA) using a SYBR-Green (Qiagen) probe. ΔΔCt values were obtained for each gene and uploaded to the Qiagen GeneGlobe online program for further analysis. Relative Gene expression was normalized against housekeeping genes on the array and genes with ≥ 5-fold increase) compared to non-treated control samples were identified. Venn diagrams to illustrate number of common genes induced by Fn-DAMPs in all three cell lines were generated by freeware (http://genevenn.sourceforge.net). Heatmaps were generated in Excel (Microsoft, Redmond, WA, USA) using relative expression levels calculated by the Qiagen GeneGlobe analysis software.

## Statistical analysis

All statistical analyses were performed using Sigma Plot version 12.0 (Systat Software, Chicago, IL, USA), with $p \leq 0.05$ considered significant. All ELISA data are presented as the mean ± standard error of the mean (SEM) of at least three independent experiments performed in triplicate. Immunoblot and WES data are representative images of one experiment repeated at least three times and data and quantifications are an average of at least three independent experiments. A Student's t-test was used to compare 2 groups and a 1-way ANOVA was used for comparing more than 2 groups with Tukey post-hoc test for multiple comparisons.

## Results

### Induction of IL-8 in fibroblasts by fibronectin DAMPS is both dose dependent and synergistic

We have previously shown that Human Embryonic Foreskin Fibroblasts (A1Fs) synthesize and secrete inflammatory cytokines such as IL-8 and TNF-α in response to the fibronectin DAMPs, FnEDA and FnIII-1c [29, 30, 57–59]. To evaluate the response of Human Adult Dermal Fibroblast (HDF) and Human Kidney Fibroblast (HKF) cell lines to fibronectin DAMPs, we treated all three cell lines with increasing doses of FnEDA and FnIII-1c for 4 h and performed ELISAs to determine IL-8 concentrations in the conditioned medium. All three cell lines showed a dose-dependent increase in IL-8 release in response to each Fn-DAMP (Fig 1A–1C). The skin fibroblasts, A1F and HDF, released similar levels of IL-8 across the dose curve (Fig 1A and 1B), while the kidney fibroblasts, HKF, synthesized relatively lower amounts of IL-8 (Fig 1C). Similar to our previous findings with A1F fibroblasts, both the HDF and HKF cells produced synergistic amounts of IL-8 when fibronectin DAMPs were added together. At 4 h, the IL-8 concentration in the conditioned medium was significantly higher than the theoretical expected additive concentration (Fig 1D–1F).

### The fibronectin DAMPs, FnEDA and FnIII-1c, activate TAK1

It has been well established that TLR signaling plays an important role in eliciting immune responses to both exogenous pathogens and products of endogenous tissue damage [60, 61]. Furthermore, this has been observed in both immune and non-immune cells [21, 61]. However, the specific mechanisms regulating this response to fibronectin DAMPs in non-immune cells are not well understood. A key component of the TLR-mediated immune response signaling is TGF-β Activated Kinase 1 (TAK1), which regulates activation of both the IKK/NF-κB and MAP kinase arms of the downstream signaling pathway [62, 63]. We, therefore, investigated whether fibronectin DAMP induced inflammatory responses were regulated by TAK1

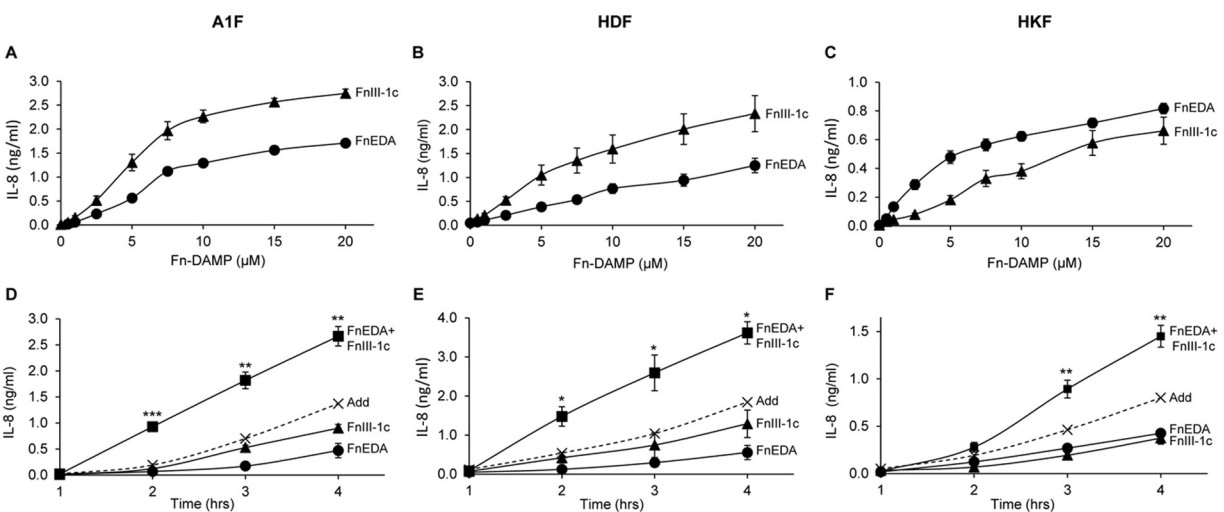

**Fig 1. Fibronectin DAMP mediated induction of IL-8 in fibroblasts.** (**A-C**) Embryonic Foreskin Fibroblasts (A1F), Human Adult Dermal Fibroblasts (HDF), and Human Adult Kidney Fibroblasts (HKF) were treated with increasing concentrations of fibronectin DAMPs, FnEDA and FnIII-1c. After 4 h, conditioned medium was collected and IL-8 concentration was determined by ELISA. (**D-F**) Cells were incubated with 5 μM FnEDA and 5 μM FnIII-1c (A1F, HDF) or 5 μM FnEDA, 10 μM FnIII-1c (HKF) individually or in combination for the indicated times. Student's t-test was used to compare the expected additive (Add) IL-8 concentration (x—x) to the actual IL-8 concentration when both DAMPs were combined (black boxes). The data represent the mean ± s.e.m. of 3 independent experiments performed in triplicate. (*P≤0.05, **P≤0.01, ***P≤0.001).

in our fibroblast cell lines by assessing phosphorylation of the Threonine 184 and 187 sites. These autophos-phorylation sites have been shown to be required for TAK1 activation [64]. Fibroblasts were treated with FnEDA or FnIII-1c alone or in combination in the presence of the TAK1 inhibitor 5Z-7-Oxozeanol (5O) or its inactive analog, 5Z-Zeanol (5Z), which served as a control. As shown in Fig 2, the immunoblots of cell lysates (Fig 2A–2C) treated with

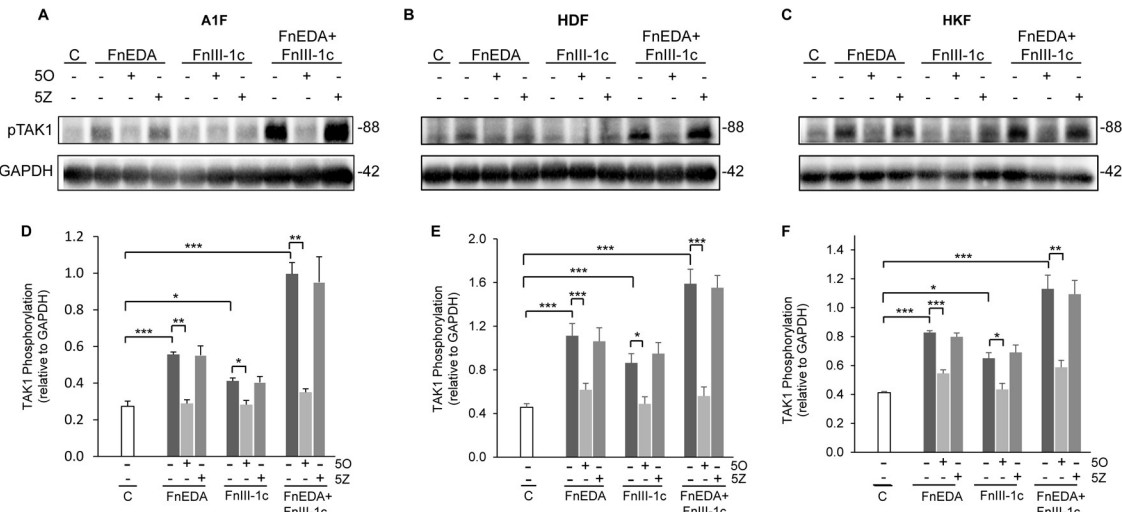

**Fig 2. TAK1 is activated in fibroblasts in response to fibronectin DAMPs.** A1F (**A,D**), HDF (**B,E**), and HKF (**C,F**) cells were treated with 1 μM of the TAK1 inhibitor 5Z-7-Oxozeaenol (5O), or the inactive analog 5Z-Zeaenol (5Z) for 1 h prior to incubation with either FnEDA (20 μM) or III-1c (20 μM), individually or in combination for an additional hour. Control cells (C) were treated with PBS/DSMO. Cells were lysed and phosphorylation of TAK1 (T184/187) was determined by western blot (**A-C**) and normalized to GAPDH (**D-F**). Blots are representative of 3 independent experiments. Data represent the mean ± s.e.m of 3 independent experiments; 1-Way ANOVA w/Tukey Post-hoc test was used for multiple comparisons. (*P≤0.05, **P≤0.01, ***P≤0.001).

FnEDA and FnIII-1c indicated an increase in TAK1 phosphorylation in all three fibroblast cell lines compared to control (C). Quantitation of these immunoblots showed that the fibronectin DAMPs significantly increased TAK1 activation which was blocked by the TAK1 inhibitor, 5O, but not the inactive analog, 5Z (Fig 2D–2F). These data are consistent with the fibronectin DAMPs inducing activation and autophosphorylation of TAK1.

## Fibronectin DAMP induced TAK activation is required for IL-8 synthesis

To determine whether TAK1 signaling regulated fibronectin DAMP mediated induction of inflammatory cytokines, we treated all three fibroblast cell lines with increasing doses of either the TAK1 inhibitor 5O or its inactive analog, 5Z for 1 h prior to addition of FnEDA and FnIII-1c either individually or in combination. After 4 h, IL-8 levels in the conditioned medium were assessed by ELISA. As shown in Fig 3, TAK1 inhibition significantly blocked fibronectin DAMP mediated IL-8 synthesis in A1F (Fig 3A–3C), HDF (Fig 3D–3F) and HKF (Fig 3G–3I) cells, while the inactive analog had no significant effect on IL-8 levels. These observations indicate that TAK1 activation is required for Fn-DAMP-mediated IL-8 release in all three fibroblast cell lines.

To further validate these findings, we utilized small interfering RNA (siRNA) mediated silencing of TAK1 expression (Fig 4). We treated A1F (Fig 4A–4C), HDF (Fig 4D–4F) and HKF (Fig 4G–4I) cells with siRNA targeting TAK1 or with a non-targeting control siRNA for 5 days followed by treatment with FnEDA and FnIII-1c either individually or together. After 4 h, IL-8 levels in the conditioned medium were evaluated by ELISA. Immunoblotting of cell lysates for TAK1 using the WES system (Fig 4A, 4D and 4G) confirmed significant siRNA-mediated reduction in TAK1 expression levels in all three cell lines (Fig 4B, 4E and 4H). Knockdown of TAK1 resulted in a significant decrease in IL-8 levels in response to fibronectin DAMPs in all three cell lines (Fig 4C, 4F and 4I). Together the data shown in Figs 3 and 4 are consistent with TAK1 kinase activity regulating the fibronectin DAMP mediated immune response in both dermal and kidney fibroblasts.

## Fn DAMP induced TAK1 activity is required for the activation of the NF-κB and MAP kinase pathways

Using the WES system, we next assessed the impact of TAK1 inhibition on the activation of the IKK/NF-κB and MAPK arms of the TLR-4 signaling pathway in all three cell lines, A1F (Fig 5), HDF (Fig 6), and HKF (Fig 7). Cells were pretreated with either the TAK1 inhibitor, 5O, or its inactive analog, 5Z, for 1 h prior to addition of FnEDA (Figs 5A, 6A and 7A) and FnIII-1c (Figs 5B, 6B and 7B) individually or in combination (Figs 5C, 6C and 7C). Addition of FnEDA and FnIII-1c individually or in combination resulted in a significant increase in the phosphorylation of IKK (Figs 5D, 6D and 7D) and NF-κB (p65) (Figs 5E, 6E and 7E) in all three cell lines.

The TAK1 dependent phosphorylation of the MAPKs, ERK, p38, and JNK, was also assessed using the WES system (Figs 5–7). Phosphorylation of ERK, p38, and JNK was increased upon treatment with FnEDA (Figs 5F, 6F and 7F) and FnIII-1c (Figs 5G, 6G and 7G) in all three cell lines. Similarly, addition of FnEDA and FnIII-1c together showed an even greater increase in phosphorylation of all three MAPKs (Figs 5H, 6H and 7H) compared to that of FnEDA and FnIII-1c individually. Additionally, treatment with the TAK1 inhibitor significantly decreased phosphorylation of the MAPKs while the inactive analog showed no inhibitory effect in any of the three cell lines (Figs 5I–5K, 6I–6K and 7I–7K). Taken together, these data are consistent with fibronectin DAMPs activating the TAK1 dependent IKK/NF-κB and MAPK arms of the TLR-4 signaling pathway to promote an inflammatory response.

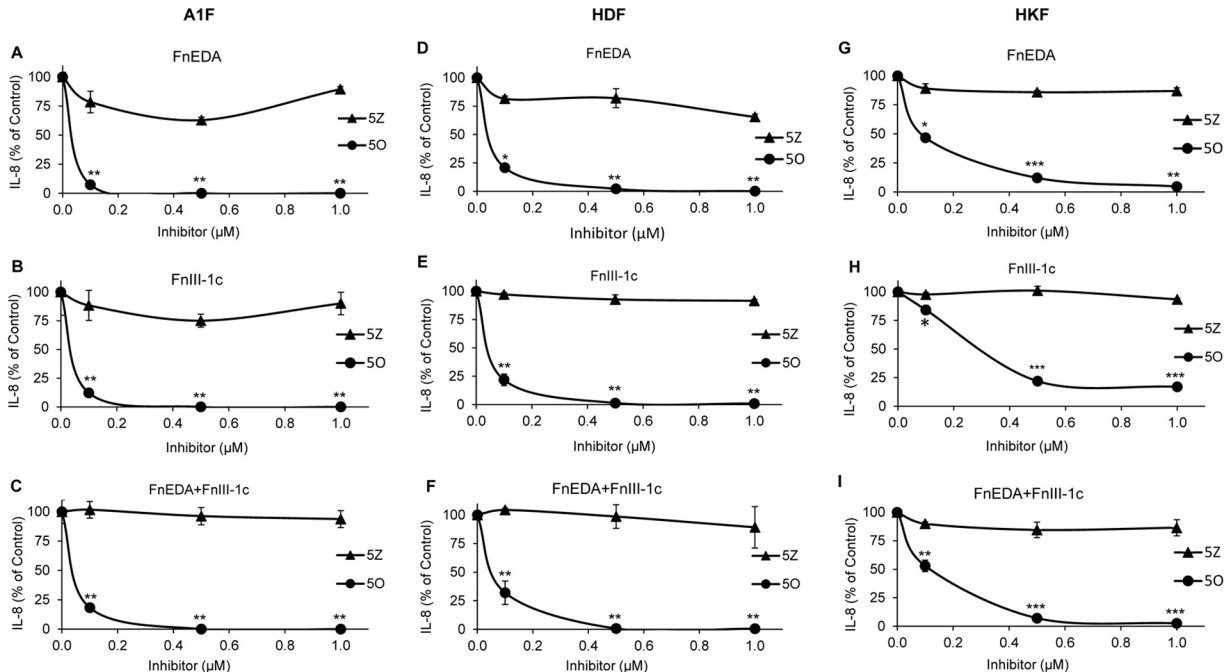

**Fig 3. TAK1 inhibitor prevents fibronectin DAMP mediated IL-8 induction.** A1F (**A-C**), HDF (**D-F**), and HKF (**G-I**) cells were treated with increasing concentrations of either the TAK1 inhibitor 5Z-7-Oxozeaenol (5O) or the inactive analog 5Z-Zeaenol (5Z) for 1 h prior to incubation with either 5 μM FnEDA or FnIII-1c (**A-F**) or 5 μM FnEDA, 10 μM FnIII-1c (**G-I**), individually or in combination, for an additional 4 h. Conditioned medium was collected and IL-8 concentration was determined by ELISA. IL-8 concentration is expressed as a percent of control (no inhibitor). The data represent the mean ± s.e.m of 3 independent experiments performed in triplicate. Student's t-test was used to compare IL-8 levels at each concentration of 5O vs 5Z. (*P≤0.05, **P≤0.01, ***P≤0.001).

## Fibronectin DAMP mediated induction of pro-inflammatory cytokines and chemokines is both, fibronectin DAMP and cell type specific

Our current findings showing that fibronectin DAMPs activate TAK1, NF-κB, and MAPKs suggest activation/utilization of similar signaling molecules/pathways in regulating the fibronectin DAMP mediated immune response in both the dermal and kidney fibroblast cell lines. However, the relative contribution of fibronectin DAMP induced TAK1 and MAPK activation in regulating the expression of specific pro-inflammatory cytokines in fibroblast cells has not been explored. To evaluate cell type and fibronectin DAMP dependent differences in the induction of inflammatory genes, we treated each of the cell lines with FnEDA (Fig 8A) or FnIII-1c (Fig 8B) for 3 h, extracted RNA and assessed changes in gene expression using an Inflammation and Autoimmunity specific (PCR) array. We found that FnEDA treatment upregulated seven genes ≥5-fold that were common to all three cell lines: TNF, IL6, CXCL8 (IL-8), CCL2, and CXCL1-3 (Fig 8A and 8C). All these genes encode chemokines and cytokines involved in the recruitment of immune cells including monocytes, neutrophils, and T-cells [65]. Additionally, more shared upregulated genes (CCL11, CCL7 and CXCL10) were seen between the two dermal fibroblast cell lines than between the dermal and kidney fibroblast cell lines (Fig 8C). Only one kidney specific gene, IL-1A, was upregulated in response to FnEDA. Interestingly, IL-1A has recently been proposed as a therapeutic target for the treatment of chronic kidney disease [66]. Similar to FnEDA, FnIII-1c also upregulated CXCL1-3 in all three cell lines (Fig 8B and 8D). In contrast to FnEDA, FnIII-1c did not induce any kidney specific genes (Fig 8D), but did induce one gene, CXCL10, specifically to the adult dermal fibroblasts. CXCL10 is thought to drive the autoimmune skin disease, vitiligo [67] as well as function as a biomarker for progression of psoriasis

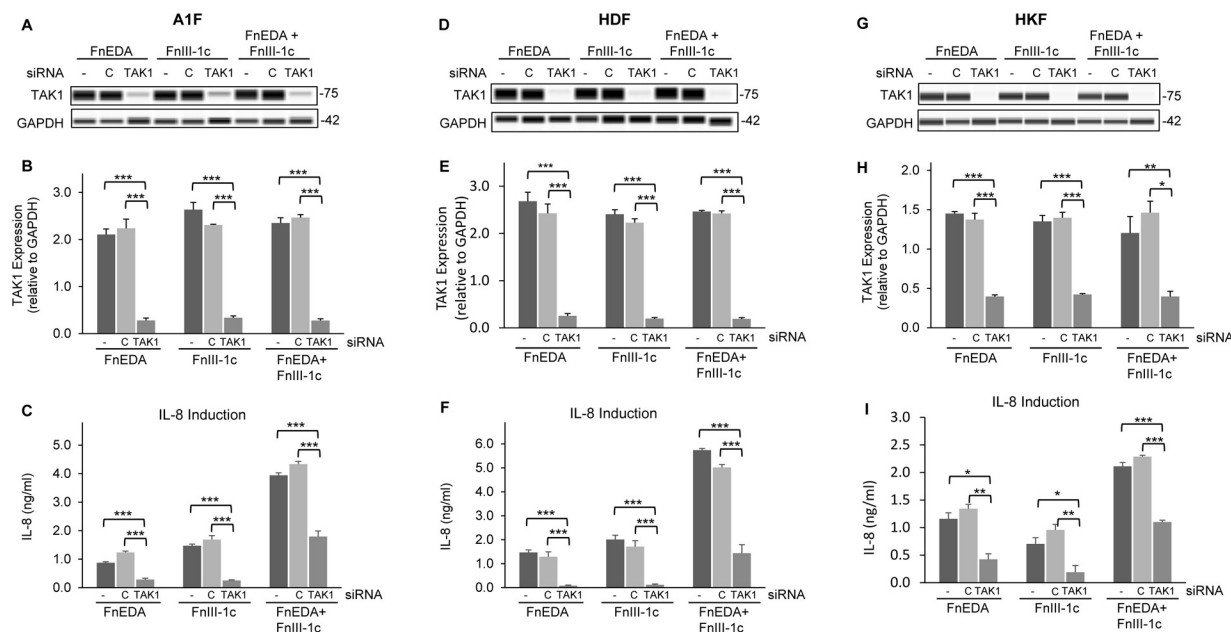

**Fig 4. siRNA-mediated knockdown of TAK1 inhibits fibronectin DAMP induced IL-8 synthesis.** A1F (**A-C**), HDF (**D-F**), and HKF (**G-I**) cells were treated with siRNA targeting TAK1 (TAK1) or non-targeting control (C) followed by treatment with either 5 μM FnEDA or FnIII-1c (**A-F**) or 5 μM FnEDA, 10 μM FnIII-1c (**G-I**), individually or in combination, for 4 h. Cells were lysed and TAK1 expression was analyzed (**A,D, G**) and quantified (**B,E,H**) by WES. Conditioned medium was collected and IL-8 concentration was determined by ELISA (**C,F,I**). The data represent the mean ± s.e.m of 3 independent experiments performed in triplicate. One Way ANOVA w/Tukey Post-hoc test was used for multiple comparisons. (*P≤0.05, **P≤0.01, ***P≤0.001).

[68]. The two dermal cell lines shared six FnIII-1c induced genes CCL2, CCL7, CXCL8, IL-6, PTGS2 and TNF (Fig 8D). Interestingly, the kidney fibroblasts were relatively less responsive to the fibronectin DAMPs compared to the dermal fibroblasts, with no genes being upregulated more than 50-fold as compared to 200-250-fold in the dermal fibroblasts (Fig 8A and 8B). These data suggest that there are organ specific differences in the inflammatory response of resident fibroblasts to fibronectin DAMPs.

To determine the relative contribution of TAK1 and the MAP Kinases in fibronectin DAMP mediated inflammatory responses, we treated all three cell lines with specific inhibitors of TAK1, ERK, JNK, or p38 prior to the addition of FnEDA or FnIII-1c. RNA was isolated and fibronectin DAMP induced genes were detected using the Inflammation and Autoimmunity PCR array. Heatmaps were generated to demonstrate the effect of inhibitor treatment on changes in FnEDA and FnIII-1c induced gene expression. We observed that the expression of all genes induced by fibronectin DAMPs was highly regulated by TAK1 in all three cell lines (Fig 9). p38 or ERK inhibition had a relatively lower impact on gene expression with most genes being only partially regulated. However, JNK inhibition reduced gene expression to a higher degree than either p38 or ERK inhibition. Taken together, our RT-PCR data suggest that 1) fibroblasts from different organs vary in their response to fibronectin DAMPs and 2) expression of fibronectin DAMP induced pro-inflammatory genes is regulated by TAK1 likely via NF-κB/JNK with less dependency on ERK and p38.

## Discussion

Remodeling of the ECM is an ongoing process in most tissues. Dysregulation of this process results in biochemical and mechanical changes in the ECM which have been shown to

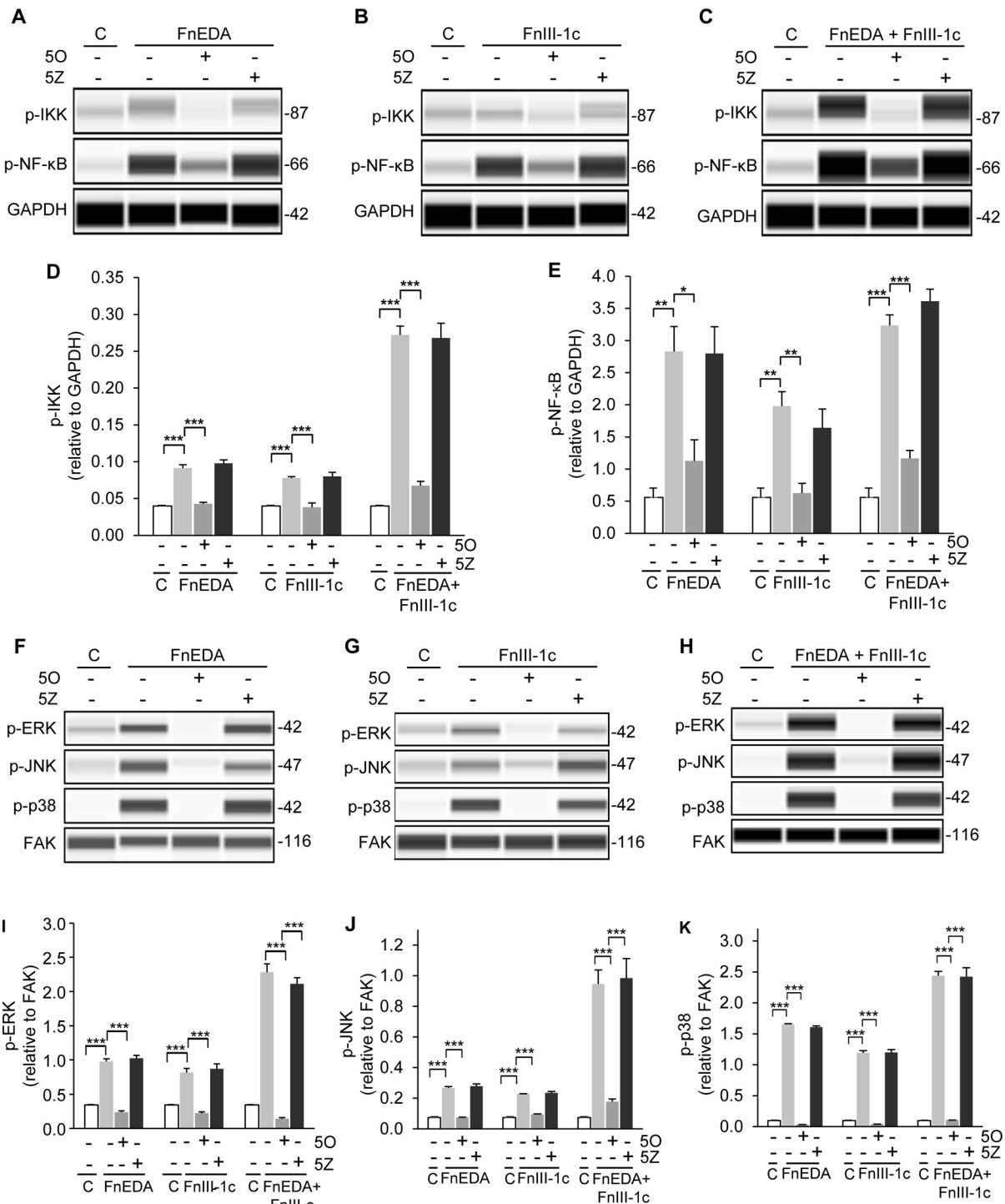

**Fig 5. Fibronectin DAMP induced activation of the NF-κB and MAPK pathways in embryonic dermal fibroblasts (A1F) requires TAK1.** A1F cells were treated with either PBS/DMSO (**C**), 1 μM of the TAK1 inhibitor (5O), or the inactive analog (5Z) for 1 h prior to incubation with FnEDA (20 μM) or FnIII-1c (20 μM), individually or in combination for an additional hour. Cells were lysed and levels of phospho-IKKα/β and phospho-NF-κB were visualized (**A-C**) and quantified using WES (**D-E**). Phosphorylation of MAPKs, ERK, JNK and p38, were visualized (**F-H**) and quantified using WES (**I-K**). Data represent the mean ± s.e.m. of 3 independent experiments. One Way ANOVA w/ Tukey Post-hoc test was used for multiple comparisons. (*P≤0.05, **P≤0.01, ***P≤0.001).

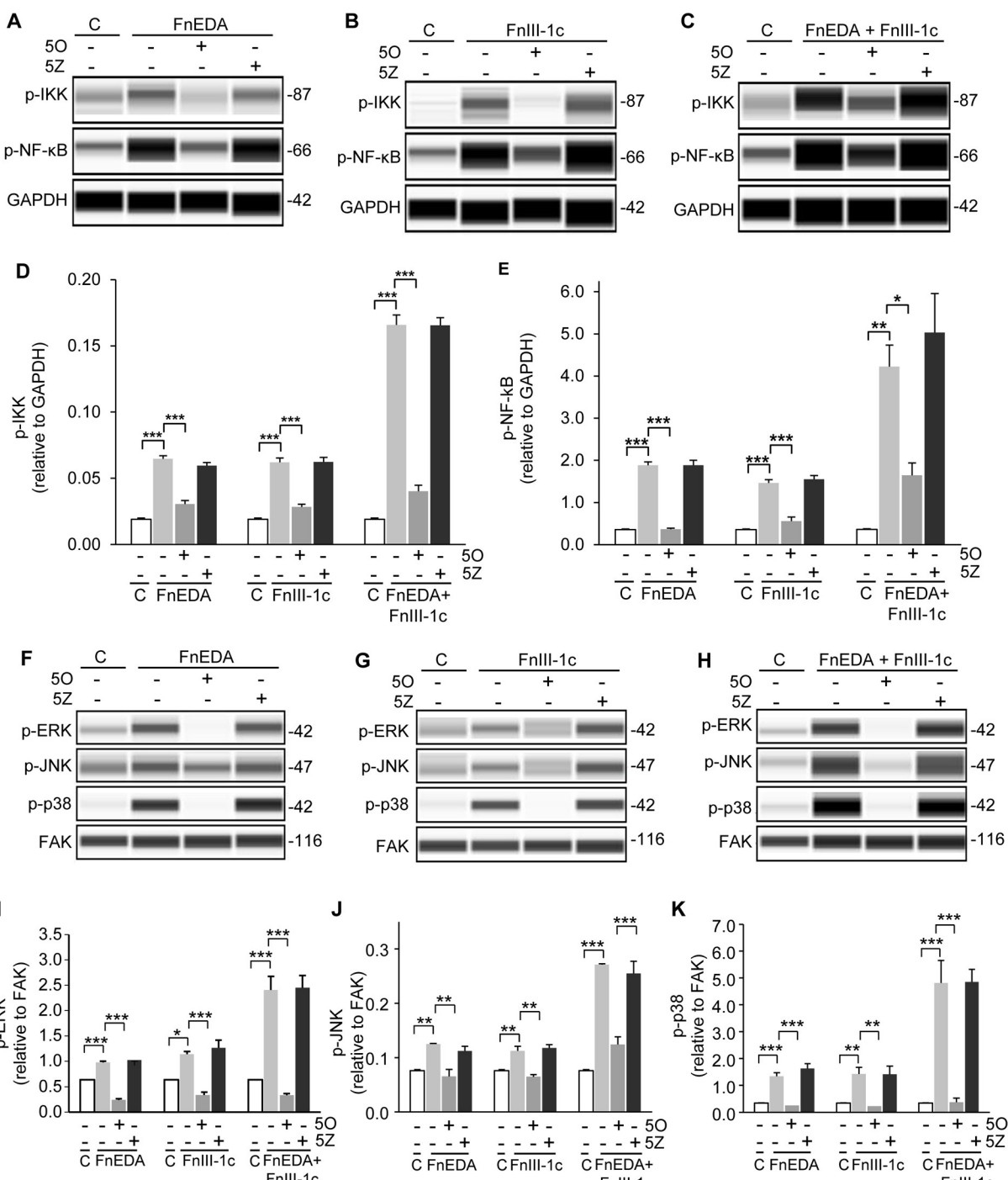

**Fig 6. Fibronectin DAMP induced activation of the NF-κB and MAPK pathways in adult dermal fibroblasts (HDF) requires TAK1.** HDF cells were treated with either PBS/DMSO (**C**), 1 μM of the TAK1 inhibitor (5O), or the inactive analog (5Z) for 1 h prior to incubation with FnEDA (20 μM) or FnIII-1c (20 μM), individually or together for an additional hour. Cells were lysed and levels of phospho-IKKα/β and phospho-NF-κB were visualized (**A-C**) and quantified WES (**D-E**). Phosphorylation of MAPKs, ERK, JNK and p38 were visualized (**F-H**) and quantified using WES (**I-K**). Data represent the mean ± s.e.m of 3 independent experiments. One Way ANOVA w/Tukey Post-hoc test was used for multiple comparisons. (*P≤0.05, **P≤0.01, ***P≤0.001).

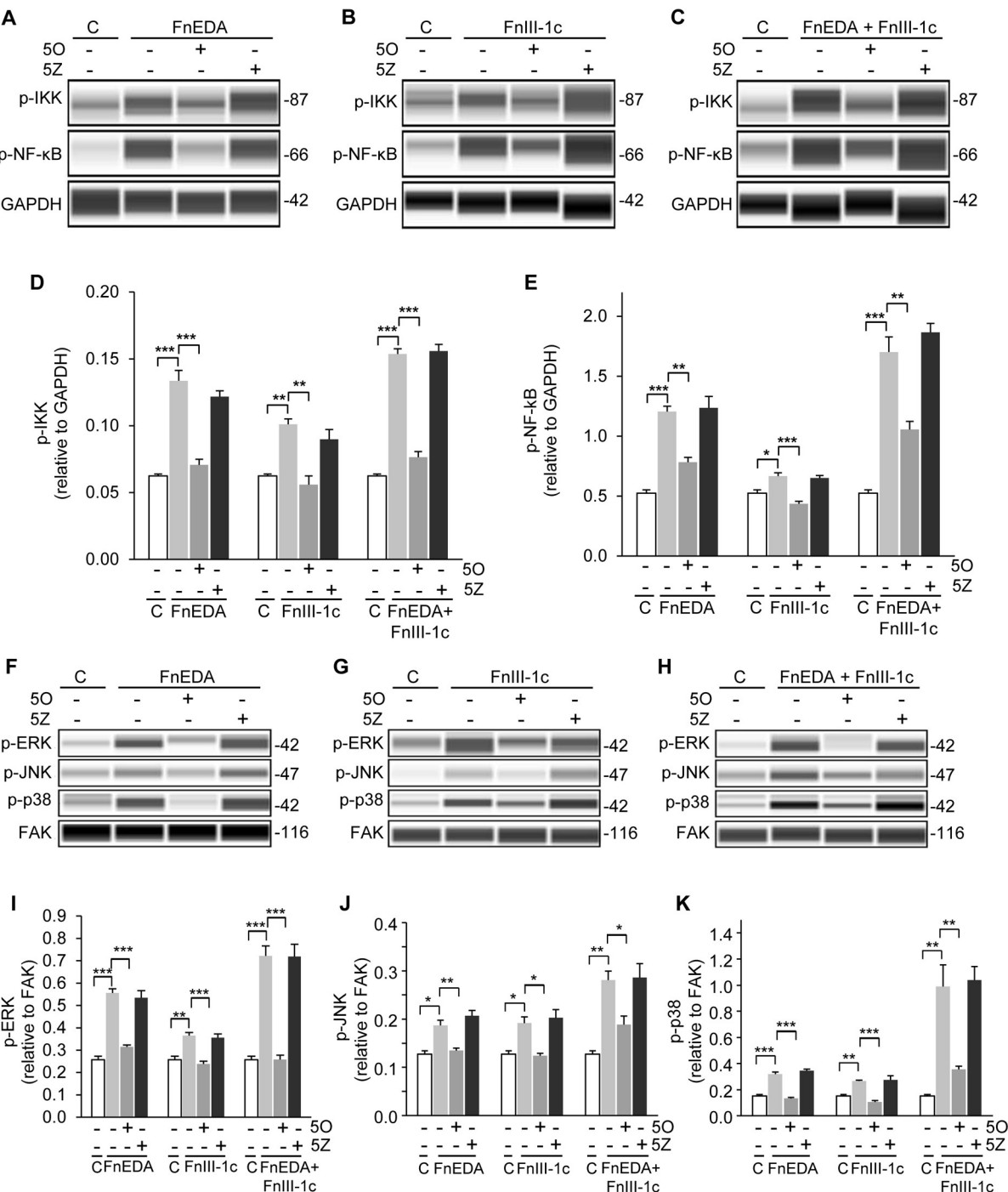

**Fig 7. Fibronectin DAMP induced activation of the NF-κB and MAPK pathways in adult kidney fibroblasts (HKF) requires TAK1.**
HKF cells were treated with PBS/DMSO which served as control (C), 1 μM of the TAK1 inhibitor (5O), or the inactive analog (5Z) for 1 h prior to incubation with FnEDA (20 μM) or FnIII-1c (20 μM), individually or in combination for an additional hour. Cells were lysed and levels of phospho-IKKα/β, phospho-NFK-κB were visualized (**A-C**) and quantified using WES (**D-E**). Phosphorylation of MAPKs, ERK, JNK and p38, were visualized (**F-H**) and quantified using WES (**I-K**). Data represent the mean ± s.e.m of 3 independent experiments. One Way ANOVA w/Tukey Post-hoc test was used for multiple comparisons. (*P≤0.05, **P≤0.01, ***P≤0.001).

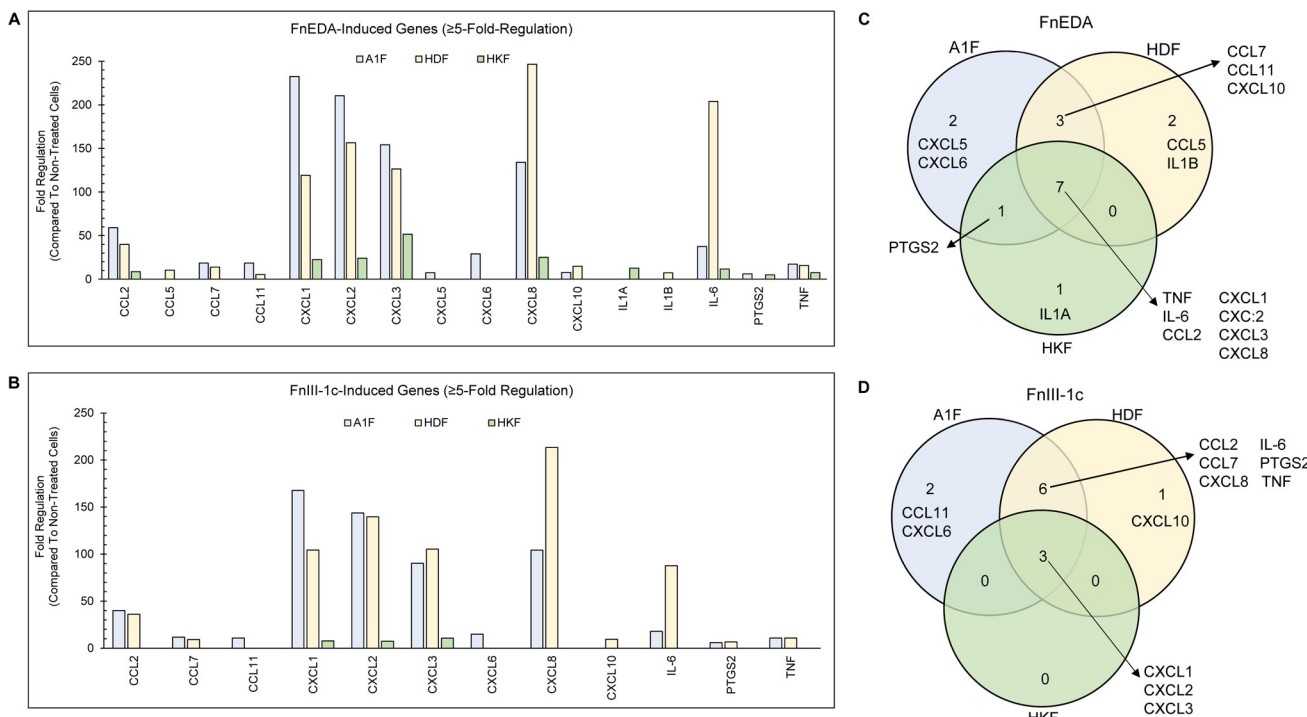

**Fig 8. Differential expression of pro-inflammatory genes in fibroblast cell lines in response to fibronectin DAMPS.** Fibroblasts were treated for 3 h with 5 μM FnEDA, 5 μM Fn-III-1c (A1F, HDF) or 5 μM FnEDA, 10 μM FnIII-1c (HKF). PBS served as control. RNA was isolated and cDNA was generated and applied to a Human Inflammatory Response and Autoimmunity PCR Array. A list of genes with ≥5-fold increase in expression compared to PBS-treated control cells was generated for FnEDA (**A**) and FnIII-1c (**B**) treated cells. The Venn-diagrams illustrate the overlap among the three cell types in the FnEDA (**C**) and FnIII-1c (**D**) induced pro-inflammatory genes. The number and names of commonly induced (in overlapping areas) and unique (inside each circle) genes are shown.

promote tissue stiffening, inflammation, and fibrosis [69]. These changes include an increase in the matrix deposition of the EDA isoform of fibronectin as well as in the mechanical unfolding of the fibronectin molecule [8]. Fibroblasts are the major cell type responsible for the synthesis, deposition, organization, and turnover of the ECM. In previous studies, we have shown that two of fibronectin's Type III domains, FnIII-1c and FnEDA, can function as DAMPs in human skin fibroblasts by serving as agonists for TLR4 [30, 33, 57]. When added to human embryonic fibroblasts these domains work individually and in synergy to activate NFκB and induce the synthesis of the fibro-inflammatory cytokines, IL-8 and TNFα [30]. The ability of these two domains to synergistically regulate the release of cytokines appears to be cell type specific as the response was not seen in THP-1 monocytes [57]. The molecular basis of the cell type specific responses to DAMPs remains poorly understood. In the current study, we compared the molecular and genetic response to fibronectin derived DAMPs in human fibroblasts from embryonic skin, adult skin and adult kidney. Fibroblasts are now recognized as major participants in the regulation of the tissue innate immune response [70]. Resident tissue fibroblasts are considered potential therapeutic targets for the treatment of chronic inflammation and fibrotic disease in both the skin [37, 71] and kidney [72].

In the current study, we show that both fibronectin derived DAMPs, FnEDA and FnIII-1c, stimulate the synthesis and release of IL-8 from all three fibroblbast cell lines and that the levels of IL-8 released are increased synergistically when the DAMPs are added together. Treatment of all three cell types with either FnEDA or FnIII-1c resulted in the activation and subsequent auto-phosphorylation of TAK1 on Thr184/187. TAK1, also known as

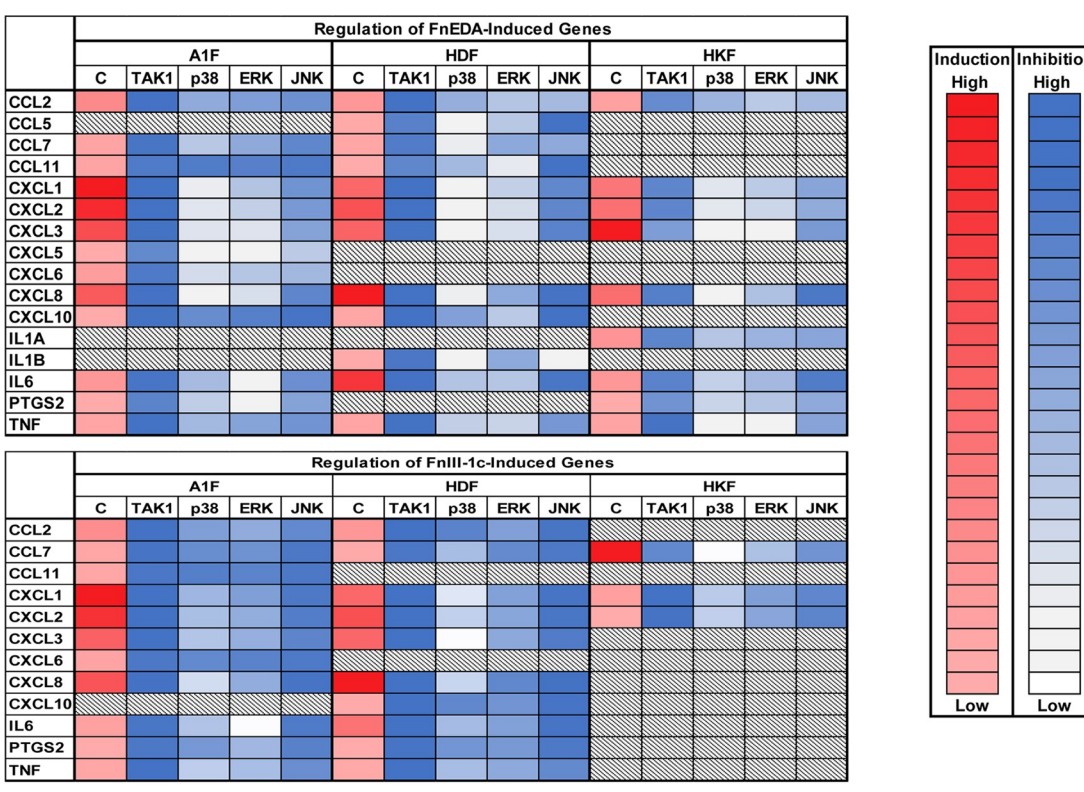

**Fig 9. The relative contribution of TAK1 and MAPK pathways in fibronectin DAMP induced expression of pro-inflammatory genes.** Fibroblast cell lines were pretreated with inhibitors to TAK1 (1 μM 5O), ERK1/2 (10 μM Temuterkib), JNK1/2/3 (10 μM JNK IX), or p38 (10 μM SB202190) for 1 h followed by addition of FnEDA (**A**) or FnIII-1c (**B**) for an additional 3 h. A1F and HDF cells received 5 μM FnEDA or 5μM FnIII-1c. The HKF cells received 5 μM FnEDA or 10 μM FnIII-1c. PBS/DMSO served as control. RNA was isolated and cDNA was generated and applied to a Human Inflammatory Response and Autoimmunity PCR Array. Fold regulation of gene expression for each cell line was determined using RT-PCR and a list of genes with ≥5-fold decrease in expression compared to control treated cells were generated. Heatmaps were generated to illustrate FnEDA (**A**) and FnIII-1c (**B**) induced genes that are regulated by TAK1 or the MAP kinases p38, ERK, and JNK. Genes not induced by fibronectin DAMPS are shown in shaded boxes.

MAP3K7, is a member of the MAPK family and known to play a major role in proinflammatory signaling initiated by a variety of fibro-inflammatory mediators including TGF-β, TNF, IL-1 and TLR ligands [63]. TAK1 activation regulates a bifurcation step in the TLR4 signaling pathway leading to the activation of the IKK/NF-κB and the MAPK arms of the pathway [63]. TAK1 phosphorylation on Thr184/187 has been shown to be required for optimal NF-κB [73] and MAPK activation [74]. Consistent with this, inhibition of TAK1 using either 5Z-7-Oxozeanol or siRNA prevented the phosphorylation of the MAPKs (ERK, JNK, and p38) and IKK/NF-κB as well as the synthesis of IL8 in response to the fibronectin DAMPs in all three cell lines. Taken together, these results indicate that the fibronectin DAMPs activated the same signaling intermediates in all three cell types.

To evaluate the inflammatory response in all three cell types, fibronectin DAMP mediated induction of inflammatory genes in the presence of specific inhibitors was analyzed using a cytokine PCR array. The findings demonstrated that TAK1 dependent cytokine induction was regulated primarily through JNK in all three cell types. JNK phosphorylates c-jun to activate the AP-1 transcription factor which is known to play a major role in the regulation of inflammatory genes in both skin and renal fibrosis [75–77]. Active JNK is elevated in diseased

kidneys and early clinical trials as well as preclinical models suggest that JNK inhibitors may have some efficacy in treating fibrotic kidney disease [78, 79]. Active JNK is also seen in keloids and hypertrophic scars of the skin where it has been proposed as a therapeutic target [80, 81]. We also observed that in addition to IL-8, both FnEDA and FnIII-1c induced the pro-inflammatory cytokines, CXCL 1,2,3 in all three cell lines. These cytokines are upregulated in psoriatic skin [82–84], allergic dermatitis [85] and in fibrotic kidney disease [86–88] and are consistent with a role for fibronectin DAMPs in the progression of skin and renal fibrosis.

In addition to common genes expressed in all three cell types, there were some cell type specific genes induced by fibronectin DAMPs. The IL-1α (IL-1A) gene was upregulated in response to FnEDA in kidney fibroblasts. Interestingly, recent studies have shown that IL-1α functions as a proinflammatory mediator and has been proposed as a therapeutic target for the treatment of chronic kidney disease [66, 89]. In adult dermal fibroblasts, FnEDA induced the CCL5 and IL-1β (IL-1B) genes while FnIII-1c induced CXCL10. CCL5 has been implicated in the progression of scleroderma [90], while IL-1β and CXCL10 have been proposed as biomarkers of disease progression in psoriasis [91, 92].

IL-6 and TNFα, which were upregulated by FnIII-1c in both skin fibroblast lines, have been implicated in the development of keloid and hypertrophic scars [93–95]. Both IL-6 and TNFα are also being evaluated in clinical trials as potential drug targets for the treatment of systemic sclerosis [96, 97]. FnIII-1c also induced expression of CCL2 in both skin fibroblast lines. CCL2 expression is regulated by c-jun in psoriatic skin [98]. Fibronectin DAMPs also increased the synthesis of CCL11 and CCL7 in skin cells. This is consistent with earlier studies showing that

CCL11 is upregulated in atopic dermatitis while CCL7 is increased in psoriasis and sclerosis where they are thought to promote skin inflammation and fibrosis [99–101].

Both hypertrophic and keloid scars arise from dysregulated wound healing which resulted in increased ECM deposition, leading to increased tissue stiffness and changes in matrix mechanics. The increase in the EDA isoform of fibronectin plays a major role in driving the changes in ECM mechanics due to its regulation of TGF-β activation [102] and the subsequent increase in myofibroblast driven contractile forces. Additionally, the binding of the α4β1 integrin to the EDA domain of fibronectin promotes a feed forward profibrotic loop in fibroblasts [11, 58]. As a consequence of the increase in tissue stiffness, the fibronectin molecules within the fibronectin fibers unfold, exposing previously cryptic sites [103].

Antibodies directed against fibronectin isoforms have been used in preclinical cancer models for targeted delivery of therapeutics including cytotoxic drugs and anti-tumor cytokines [104]. Similarly, fibronectin binding peptides which selectively recognize different conformations of fibronectin are being used to distinguish relaxed from stretched fibers. Peptides which bind to stretched fibers have been shown to accumulate in tumors and fibrotic areas, allowing for targeted delivery of therapeutic agents thus preventing unwanted systemic side effects [105, 106]. Developing reagents to specifically target biochemical or mechanical changes in ECM fibronectin may provide novel approaches to the treatment of diseases driven by fibro-inflammation.

## Supporting information

**S1 Dataset.**
(PDF)

**S1 Raw images.**
(PDF)

## Author Contributions

**Conceptualization:** Paula J. McKeown-Longo.

**Data curation:** Pranav Maddali.

**Formal analysis:** Pranav Maddali, Anthony Ambesi, Paula J. McKeown-Longo.

**Funding acquisition:** Paula J. McKeown-Longo.

**Methodology:** Pranav Maddali, Anthony Ambesi.

**Project administration:** Paula J. McKeown-Longo.

**Supervision:** Anthony Ambesi, Paula J. McKeown-Longo.

**Writing – original draft:** Pranav Maddali.

**Writing – review & editing:** Paula J. McKeown-Longo.

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
