## [Decision Letter · Decision Letter 0]

28 Feb 2023

PONE-D-23-02446Induction of Pro-Inflammatory Genes by Fibronectin DAMPS in Three Fibroblast Cell Lines:  Role of TAK1 and MAP kinasesPLOS ONE

Dear Dr. McKeown-Longo,

Thank you for submitting your manuscript to PLOS ONE. After careful consideration, we feel that it has merit but does not fully meet PLOS ONE’s publication criteria as it currently stands. Therefore, we invite you to submit a revised version of the manuscript that addresses the points raised during the review process.

We look forward to receiving your revised manuscript.

Kind regards,

Nazmul Haque

Academic Editor

PLOS ONE

Journal Requirements:

"Burke Biomedical Research Foundation

Muntz Fund"

"No authors have competing interests"

Reviewers' comments:

Reviewer's Responses to Questions

**Comments to the Author**

1. Is the manuscript technically sound, and do the data support the conclusions?

Reviewer #1: Yes

Reviewer #2: Yes

2. Has the statistical analysis been performed appropriately and rigorously? 

Reviewer #1: Yes

Reviewer #2: Yes

3. Have the authors made all data underlying the findings in their manuscript fully available?

Reviewer #1: Yes

Reviewer #2: Yes

4. Is the manuscript presented in an intelligible fashion and written in standard English?

Reviewer #1: Yes

Reviewer #2: Yes

5. Review Comments to the Author

Reviewer #1: This is a rather straightforward study by experienced groups that address the remodeling of the extracellular matrix (ECM) and the Toll- Like Receptor-4 (TLR4) complexes and downstream signaling pathways in chronic inflammation. The authors previously showed that fibronectin’s extra domain A (FnEDA) and the partially unfolded first Type III domain (FnIII-1c) function as Damage Associated Molecular Pattern (DAMP) molecules to stimulate the induction of inflammatory cytokines by serving as agonists for TLR4. In this manuscript, they investigated the molecular steps regulating the fibronectin driven induction of inflammatory genes in three human fibroblast cell lines (human foreskin fibroblasts: A1F, adult human dermal fibroblasts: HDF, and Human Kidney fibroblasts: HKF).

Using the inhibitor and siRNA-mediated gene silencing of TAK1 and the automated capillary-based western system (WES), they showed that

1) Fibronectin DAMP mediates induction of IL-8 in fibroblasts (figure 1).

2) Fibronectin DAMPs, FnEDA and FnIII-1c, phosphorylate TAK1 at Thr184/187 in fibroblasts (figure 2).

3) TAK1 kinase activity regulating the fibronectin DAMP mediates IL-8 synthesis in both dermal and kidney fibroblasts (figure 3 and 4).

4) Fibronectin DAMP induced activation of the NF-κB and MAPK pathways requires TAK1 in all three cell lines (figure 5-7).

Using RT² Profiler PCR Array, they further demonstrated that

5) that differential expression of pro-inflammatory genes in fibroblast cell lines in response to fibronectin DAMPS (figure 8).

6) JNK inhibition reduced gene expression induced by fibronectin DAMPs/TAK1 to a higher degree than either p38 or ERK inhibition (figure 9).

Based on the above results they propose that FnEDA and FnIII-1c induce several pro-inflammatory cytokines whose expression is dependent on both TAK1 and JNK MAPK and highlight cell-type specific differences in the gene-expression profiles of the fibroblast cell-lines. I think this work is well organized and a nice contribution to the field of the remodeling of the ECM. This work could serve as a basis for further consideration of signaling molecules downstream of fibronectin DAMPs as a potential therapeutic target for hypertrophic skin disease.

I have only a concern to address as below.

There are no blot data in figure 6F, 6G, and 7F.

Reviewer #2: In this study by Maddali et al., the authors explored the role of two fibronectin DAMPs (FnEDA and FnIII-1c) in the activation of TGFb activated kinase 1 (TAK1) in fibroblasts derived from the kidney, foreskin or dermis. The authors show that both FnEDA and FnIII-1c are capable of activating the TAK1 and MAPK pathways in all three cell types, but that interestingly there appears to be organ specific differences in the inflammatory response. In particular they showed that with the exception of IL-8 different pro-inflammatory cytokines are upregulated in the different cell types. The authors also show that activation of TAK1 leads to activation of the IKK/NF-kB and MAPK pathways. This is a well written and very interesting study.

One point that should be addressed, however, is the observation that FnEDA and FnIII-1c appear to vary in their ability to IKK/NF-kB and MAPK pathways (Figure). FnIII-1c seems to be a weaker DAMP then FnEDA and did not show a very robust response. Do the authors have any explanation why this would be the case?

6. PLOS authors have the option to publish the peer review history of their article (what does this mean?). If published, this will include your full peer review and any attached files.

Reviewer #1: No

Reviewer #2: No

---

## [Author Response · Author response to Decision Letter 0]

20 Apr 2023

Response to Reviewers:

Response to Reviewer 1:

There are no blot data in Figures 6F, 6G and 7F. These data are now included. 

Response to reviewer 2

FnIII-1c appears to be a weaker DAMP than FnEDA and did not show a very robust response. Do the authors have any explanation why this would be the case?

 This is an interesting question. We do not know the underlying mechanism of how these DAMPs activate TLR signaling. Presumably activation of TLR4 by Fn DAMPs may involve ancillary proteins or adaptor molecules as have been described for LPS. Identification of such molecules and their role in regulating downstream signaling may provide further insight into this question.

---

## [Decision Letter · Decision Letter 1]

15 May 2023

Induction of Pro-Inflammatory Genes by Fibronectin DAMPS in Three Fibroblast Cell Lines:  Role of TAK1 and MAP kinases

PONE-D-23-02446R1

Dear Dr. McKeown-Longo,

We’re pleased to inform you that your manuscript has been judged scientifically suitable for publication and will be formally accepted for publication once it meets all outstanding technical requirements.

Kind regards,

Nazmul Haque

Academic Editor

PLOS ONE

Additional Editor Comments (optional):

Reviewers' comments:

Reviewer's Responses to Questions

**Comments to the Author**

1. If the authors have adequately addressed your comments raised in a previous round of review and you feel that this manuscript is now acceptable for publication, you may indicate that here to bypass the “Comments to the Author” section, enter your conflict of interest statement in the “Confidential to Editor” section, and submit your "Accept" recommendation.

Reviewer #1: All comments have been addressed

Reviewer #2: All comments have been addressed

2. Is the manuscript technically sound, and do the data support the conclusions?

Reviewer #1: Yes

Reviewer #2: Yes

3. Has the statistical analysis been performed appropriately and rigorously? 

Reviewer #1: Yes

Reviewer #2: Yes

4. Have the authors made all data underlying the findings in their manuscript fully available?

Reviewer #1: Yes

Reviewer #2: Yes

5. Is the manuscript presented in an intelligible fashion and written in standard English?

Reviewer #1: Yes

Reviewer #2: Yes

6. Review Comments to the Author

Reviewer #1: In my opinion, in this revised manuscript, the responses to the major points raised previously were adequate and now this is acceptable for publication in PLOS ONE.

Reviewer #2: (No Response)

7. PLOS authors have the option to publish the peer review history of their article (what does this mean?). If published, this will include your full peer review and any attached files.

Reviewer #1: No

Reviewer #2: No

---

## [Editor Report · Acceptance letter]

17 May 2023

PONE-D-23-02446R1 

Induction of Pro-inflammatory Genes by Fibronectin DAMPs in Three Fibroblast Cell Lines: Role of TAK1 and MAP Kinases 

Dear Dr. McKeown-Longo:

I'm pleased to inform you that your manuscript has been deemed suitable for publication in PLOS ONE. Congratulations! Your manuscript is now with our production department. 

Kind regards, 

on behalf of

Dr. Nazmul Haque 

Academic Editor

PLOS ONE